# Recurrent Hemorrhagic Conversion of Ischemic Stroke in a Patient with Mechanical Heart Valve: A Case Report and Literature Review

**DOI:** 10.3390/brainsci8010012

**Published:** 2018-01-07

**Authors:** Micheal Jace Tarver, Tyler Schmidt, Michael T. Koltz

**Affiliations:** Division of Neurosurgery, Department of Surgery and Perioperative Care, Dell Medical School, University of Texas at Austin, Austin, TX 78712, USA; micheal.j.tarver@utexas.edu (M.J.T.); tyler.schmidt@utexas.edu (T.S.)

**Keywords:** hemorrhagic conversion, ischemic stroke, St. Jude mechanical heart valve, anticoagulation, Magnum heart valve

## Abstract

The authors present a unique case of recurrent stroke, discovered to be secondary to hemorrhagic conversion of microemboli from a mechanical aortic valve despite anticoagulation with Coumadin. The complexity of this case was magnified by the patient’s young age, a mechanical heart valve (MHV), and a need for anticoagulation to maintain MHV patency in a setting of potentially life-threatening intracranial hemorrhage. Anticoagulant and antiplatelet therapy are risk factors for hemorrhagic conversion post-cerebral ischemia; however, the pathophysiology underlying endothelial cell dysfunction causing red blood cell extravasation is an active area of basic and clinical research. The need for randomized clinical trials to aid in the creation of standardized treatment protocol continues to go unmet. Consequently, there is marked variation in therapeutic approaches to treating intracranial hemorrhage in patients with an MHV. Unfortunately, patients with an MHV are considered at high thromboembolic (TE) risk, and these patients are often excluded from clinical trials of acute stroke due to their increased TE potential. The authors feel this case represents an example of endothelial dysfunction secondary to microthrombotic events originating from an MHV, which caused ischemic stroke with hemorrhagic conversion complicated by the need for anticoagulation for an MHV. This case offers a definitive treatment algorithm for a complex clinical dilemma.

## 1. Introduction

Stroke is the fifth most common cause of death in the U.S. and is the leading cause of long-term physical and cognitive impairment, with over 800,000 stroke incidents occurring annually in the United States [1,2,3,4]. The risk of stroke secondary to thromboembolism is significantly increased with the use of a mechanical heart valve (MHV) as compared to bioprosthetic valves [5,6]. Consequently, MHVs require patients to undergo lifelong anticoagulation. Major thromboembolic (TE) events that result in death or deficit, or necessitate surgery, have an increased incidence in patients with an MHV that are without antithrombotic treatment, as compared to patients that receive antithrombotic therapy. The incidence of major TE events for patients with an MHV without TE therapy, with antiplatelet therapy, and with warfarin are, respectively, 4, 2.2, and 1 per 100 patient-years [7]. A stark contrast is seen between the incidence of hemorrhagic stroke in patients taking Warfarin and the general population, 2–3 per 100 patient-years and 25 per 100,000, respectively [8,9,10]. The formation of a thrombus on the surface of the MHV can cause malfunctions in the valve’s function. However, danger also presents itself when a fragment of the thrombus is dislodged from the valvular surface and travels through the circulation. This fragment can become lodged in vasculature and thus occlude the vessel, resulting in ischemia and consequently hypoxia and infarction. One of the most common sites for thrombus to embolize is the cerebral cortex due to the disproportionately high blood flow compared to other regions of the cerebrum. The occlusion of vasculature can lead to the metabolic disruption of the cerebral capillaries and eventually to hemorrhagic conversion in a step-wise process that ultimately ends in endothelial cell integrity failure and extravasation of vascular contents into the parenchyma, as described by Simard et al. [11].

Therapeutic recommendations are currently based on an individualized assessment of the risk–benefit ratio, as no established guidelines exist. Overall, the goal of the patient-centered recommendation is to prevent recurrent ischemic events without causing new or worsening hemorrhage. A recent cross-sectional study involving neurosurgeons and thrombosis specialists highlighted the continued need for randomized clinical studies citing marked variation in their therapeutic approaches to treating intracranial hemorrhage in patients with an MHV [12].

## 2. Case Presentation

A 43-year-old Vietnamese man, with no history of hypertension, currently living in the US, first presented with a five-day history of persistent worsening headache and dizziness with intermittent nausea. He is a Type II diabetic with a significant medical history of a mechanical aortic valve replacement (St. Jude valve placed in Vietnam in 2008) requiring anticoagulation treatment with Coumadin. A routine workup included a non-contrast head CT that showed a 17 mm × 17 mm hypodense ovoid lesion in the cerebellar vermis with minimal effacement of the left wall of the fourth ventricle (Figure 1A). Labs at presentation showed an International normalized ratio (INR) of 5.66. Diagnostic Magentic Resonance Imaging (MRI) showed these to be hemorrhagic lesions favored to be multiple cavernous hemangiomas rather than metastatic or septic emboli. This conclusion was also supported by a transthoracic echocardiogram done at this admission which showed a 70% LVEF with no evidence of thrombosis. Coumadin was temporarily discontinued, and INR therefore decreased to subtherapeutic 1.42 prior to discharge. Coumadin therapy was restarted on Post-Bleed Day 7.

The patient returned to the Emergency Department (ED) less than a month later reporting a one-day history of headache. Head CT showed a 3 cm × 5 cm left parietal intracranial hemorrhage (Figure 1B). Risks and benefits of medical therapy and surgical intervention were considered by a multidisciplinary team. Cardiology advised against holding anticoagulation for an extended time due to a risk of thrombosis with an MHV. Although the St. Jude valve is a high-flow bi-leaflet MHV with a relatively low-risk of thrombosis, the elevated risk of acute valvular thrombosis with corresponding risk for acute heart failure from leaflet fixation or sudden death deemed anticoagulant discontinuation unadvisable. The feasibility of replacing the mechanical aortic valve with a bioprosthesis to allow for permanent discontinuation of anticoagulation therapy was discussed at this time; however, it was not recommended due to the patient’s relatively young age, the associated morbidity of valvular redo surgery, and the decreased lifespan of bioprosthetic valves relative to MHV. Surgical evacuation was recommended to the patient, and a left temporal craniotomy for evacuation of the intraparenchymal hematoma was performed. The pathology report of the excised hematoma showed only a “blood clot with scant fragments of brain parenchyma with ischemic changes” and thus was not diagnostic for vascular malformation.

Three months later, the patient returned to the ED due to acute onset worsening left-sided weakness and numbness that started at lunchtime and continued to worsen into the night. A head CT showed acute right parietal intracranial hemorrhage with associated subarachnoid extension (Figure 1C).

The multidisciplinary team was again consulted for this third episode of recurrent intracranial hemorrhage. Coumadin was reversed and ASA 81 mg daily was started on Post-Bleed Day 7. Gradient Recalled Echo (GRE) T2-weighted 3T MRI showed innumerable bilateral, multilobar hypointensities, indicating micro-hemorrhages associated with embolic ischemic events. To monitor anticoagulation, prothrombin time (PT), partial thromboplastin time (PTT), and INR were drawn and measured regularly, showing subtherapeutic to low-normal INR levels (1.1–2.8) consistent with previous measurements since the first hemorrhagic event three months prior. A hypercoagulability panel was performed and showed no signs of hypercoagulability, including normal assay results for von Willebrand factor and factors V, VIII, X, and lupus anticoagulant was not detected. PT was elevated, and phospholipid-dependent screening tests (PT, Dilute Russells viper venom time (DRVVT)) were not prolonged. In addition, patients underwent EKG and telemetry (as he had for each previous admission) with normal results, suggesting low likelihood of emboli from cardiac thrombosis. These findings favored ongoing cerebral emboli over hemorrhagic metastases, cavernomas, infection, or other considerations (Figure 2). Transfemoral angiography definitively ruled out aneurysm, arteriovenous malformation, arteriovenous fistula, and other vascular anomaly as potential sources of hemorrhage. Cardiology recommended replacing the mechanical St. Jude valve with a bioprosthesis. Cardiothoracic surgery then performed a redo sternotomy with an aortic valve replacement using a 27 mm Magna tissue valve four weeks after this third admission. The patient was treated post-operatively with aspirin and Plavix for 6 months, after which antiplatelet therapy may be discontinued indefinitely. The patient recovered well post-operatively with no complications. His neurologic function returned to normal by the sixth week after operative appointment.

## 3. Discussion

Reperfusion of ischemic cerebral vasculature is known to cause vascular endothelial dysfunction that can result in the formation of edema and hemorrhagic conversion of the ischemic insult. The intricate process by which this occurs is thought be an amalgam of cellular and molecular processes the exact mechanisms of which are currently being studied [11,13].

A step-wise view of the cause of hemorrhagic conversion following cerebral ischemia posits that the hypoxic conditions, with or without free radical formation following reperfusion, can lead to a disruption of cellular functioning initiating a cascade of effects leading to disruption of the blood–brain barrier. This step-wise process begins with ischemia-induced decreases in intracellular ATP concentrations that promote ionic edema, leading to Starling force-driven vasogenic edema [11]. The vasogenic edema is followed by endothelial cell death and, consequently, intercellular structural failure, causing hemorrhagic conversion via extravasation of vascular contents into the parenchymal surroundings upon reperfusion [14,15].

An inflammatory extracellular matrix (ECM) degradation surrounding the infarct during this process is thought to be mediated via aberrant functioning or upregulated matrix metalloproteinases (MMPs) [13,16,17]. MMP-2 and MMP-9 have been shown to play a significant role in hemorrhagic conversion and the reduction of their activity in the initial stages of cerebral ischemia corresponds to a decrease in the incidence of hemorrhagic conversion [18].

In 1999, Simard et al. stratified TE potential into high-, middle-, and low-risk categories and recommended therapeutic strategies for the cessation and reinstatement of anticoagulants within these risk categories for those undergoing intracranial procedures [19]. Cerebrovascular accident secondary to atrial fibrillation or intracardiac thrombus are categorized as high-risk and are cases in which surgery should be avoided unless necessary. Reinstitution of anticoagulants such as non-bolus heparin post-operatively and oral anticoagulants can resume 3–5 days after surgery in these high-risk patients [19]. Patients with an MHV, valvular disease, and atrial fibrillation, atrial fibrillation 2–12 months after cerebrovascular accident fall into the moderate-risk category. These moderate-risk patients should receive post-operative low-dose subcutaneous heparin and start oral anticoagulation between Days 5 and 7 post-operation. The low-risk group includes patients with bioprosthetic valves should receive post-operative low-dose subcutaneous heparin and oral anticoagulation should begin 7–14 days post-operatively [19]. The recommendations represent the need to balance the risks of hemorrhage and anticoagulation in relation to the wound healing process to minimize negative outcomes. Although these recommendations are nicely packaged, the authors heavily underscore the above recommendations are simply clinical options for treatment rather than set-in-stone therapeutic guidelines and that clinical judgement and experience ought to take precedence. This relegation is due to the scarcity of randomized clinical trials for which the authors hope will eventually be carried out to establish efficacious therapeutic guidelines [19]. This need for randomized clinical trials continues as highlighted by a 2014 cross-sectional study involving neurosurgeons and thrombosis specialists citing marked variation in their therapeutic approaches to treating intracranial hemorrhage in patients with an MHV [12].

In the presented case, the multi-disciplinary team withheld anticoagulants until the hemorrhage was stable on head CT and seven days had passed from the sentinel event. Aspirin 81 mg daily was then started for another 14 days to prevent exacerbation of the hemorrhage. Repeat MRI confirms the unique aspect of this case by potentially contributing to a pathologic mechanism for this disease process. Specifically, the preoperative GRE T2-weighted 3-Tesla MRI demonstrated several multilobar, bihemispheric hypointense lesions in multiple vascular territories, providing strong evidence for a microembolic pathophysiology that is most likely explained by the presence of a foreign mechanical valve. The hypointense regions of hemorrhage seen on GRE T2-weighted 3-Tesla MRI imaging correspond directly to the contrast-enhancing areas on T1 post-contrast imaging with gadolinium (Figure 2B). This congruity suggests endothelial cell dysfunction and proposes a mechanism to explain the observed hemorrhage. GRE hypointensities represent focal ischemia and hemorrhage, whereas contrast-enhancing regions on T1 post-contrast imaging represent membrane permeability with gadolinium extravasation into the extracellular space. Since these hypointensities and contrast-enhancing regions are congruent, it is reasonable to conclude that ischemia was followed by endothelial cell dysfunction, increased vascular membrane permeability, and extravasation of red blood cells into the parenchyma, as seen by the hypointense blossoming on the GRE (Figure 2). This hypothesis is supported by the tissue examined by the pathologist at the time of craniotomy: hemorrhage with ischemic changes. Given the clinical course, MRI findings, the presence of a mechanical aortic valve, and intermittent holding of anticoagulation with three months of subtherapeutic/low normal INR values in this patient, the hypothesis of microemboli forming on and spreading from the MHV is well supported.

Much of the literature suggest that anticoagulant and antiplatelet therapies are a risk factor for hemorrhagic conversion along with other risk factors such as renal impairment, hyperglycemia, and advanced age [19,20,21,22,23,24]. Anticoagulant therapy has also been shown to increase the risk of hemorrhagic conversion post-cerebral ischemia in the absence of tissue plasminogen activator (tPA) administration in studies using various anticoagulants, including warfarin [25,26]. However, a retrospective analysis study suggest that anticoagulative therapy alone does not increase the risk of hemorrhagic conversion but does increase the severity of hemorrhagic conversion when it occurs. The ability of this study to resolve anticoagulation as a modifiable risk factor in hemorrhagic conversion was hindered due to much of the study population being on anticoagulants [27].

Heart valve replacement has been associated with cerebral ischemia. Thromboembolisms in patients with valve replacements have been approximated to be 1% per patient-year with cerebrovascular event history recognized as a risk factor for TE and hemorrhagic complications [28,29,30,31]. As mentioned above, anticoagulation decreases the risk of TE in patients with an MHV; however, the risk of thromboembolism is significantly increased with the use of an MHV as compared to bioprosthetic valves. Although warfarin is efficacious at reducing risk of TE in patients with an MHV, warfarin heightens hemorrhagic risk, as therapeutic levels are difficult to achieve and maintain [32,33,34,35,36].

Of the three main classes of MHV (tilting-disk, caged-ball, and bi-leaflet valves), the bi-leaflet implants have the least potential for thromboembolism formation relative to caged ball valves, which have the highest potential. The differences in embolism formation potential are due to differences in material properties and mechanical actuation unique to the implanted valves. TE potential is also related to the position of the valve in the heart irrespective of valve type [5,6]. Site-specific variation in hemodynamics modulate the risk of thromboembolism of aortic and mitral valve replacements such that aortic valves, compared to mitral valve replacements, have a decreased TE potential [37,38].

MHVs have about twice the lifespan of tissue-derived bioprosthetic valves and thus have lower re-operation rates. As such, a mechanical valve replacement is typically used in patients below the age of 60 since the morbidity associated with re-operation is greater than that of the TE and hemorrhagic risks [7,32,33,34,39,40,41,42]. Although patients with an MHV are considered at high TE risk, little data on the use of anticoagulation after ischemic stroke is available, as these patients are often excluded from clinical trials of acute stroke due to their increased TE potential [6,43,44].

## 4. Conclusions

The authors feel the presented case offers an opportunity to support an existing proposed mechanism for hemorrhagic conversion of ischemic stroke and suggests a possible treatment pathway for a complex clinical dilemma: recurrent hemorrhagic stroke in a patient “needing” anticoagulation. This is a unique case, as it gives rise to a histopathologic and radiographic comparison that supports the hypothesis of microemboli causing focal ischemia with likely endothelial cell dysfunction and ultimately hemorrhagic stroke. Patients with an MHV on anticoagulants that have sustained a hemorrhagic stroke demonstrate the delicate counter-balance of risk associated with management of such variables. Tissue heart valves that allow for anti-platelet therapy may be a safer option in patients prone to hemorrhage with anticoagulants.

## Figures and Tables

**Figure 1 brainsci-08-00012-f001:**
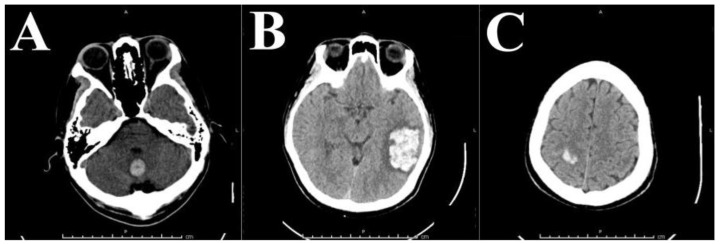
Serial head CT without contrast over 3 month period. (**A**) Initial head CT without contrast showing midline hyperdensity in the cerebellar vermis likely representing intraparenchymal hematoma without hydrocephalus. (**B**) Head CT without contrast one month after initial visit showing large left temporoparietal hyperdensity. (**C**) Head CT without contrast three months after initial visit showing right parietal hyperdensity.

**Figure 2 brainsci-08-00012-f002:**
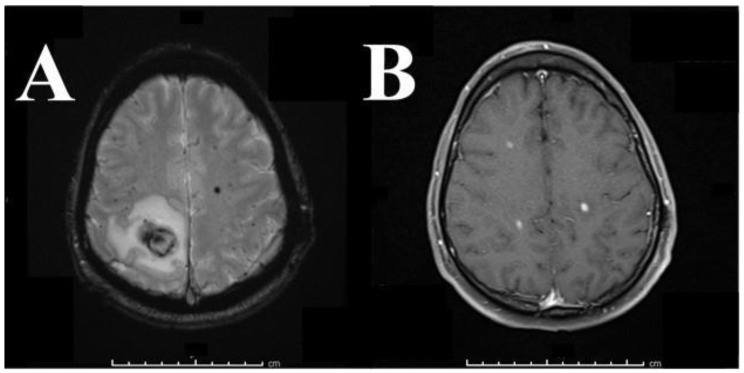
(**A**) 3T T2-weighted GRE MRI 3 months from initial presentation. Representative image shows bihemispheric multi-lobar hypointense lesions, indicating blossoming of microhemorrhage likely from thrombotic events from the mechanical heart valve. The patient did have a left temporal lesion removed during his second hospital stay, with surgical pathology showing hemorrhage without underlying lesion. (**B**) T1-weighted, post-contrast 3-Tesla MRI showing multiple enhancing lesions. (A,B) in combination are highly suggestive of endothelial cell dysfunction with extraluminal hemosiderin and contrast deposition.

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
