# Peer review of "Recurrent Hemorrhagic Conversion of Ischemic Stroke in a Patient with Mechanical Heart Valve: A Case Report and Literature Review"

_brainsci, 2018, doi:10.3390/brainsci8010012_

Round 1
Reviewer 1 Report
The paper entitled " Management of Recurrent Hemorrhagic Conversion of Ischemic Stroke
Caused By Anticoagulation for Mechanical Heart Valve" suggests that this case offers an opportunity to support an existing proposed mechanism for hemorrhagic conversion of ischemic stroke.
The study is well-designed and argumentation is persuasive.
Only 2 minor modifications should be applied:
Authors should add the abbreviation of ECM (extracellular matrix) and tPA (tissue Plasminogen Activator).
Author Response
Dear Sir or Madam,
Thank you for taking the time to review our case report.
We have added "ECM (extracellular matrix) and tPA (tissue Plasminogen Activator)" to the abbreviation section as requested.
Sincerely,
Michael T. Koltz
Reviewer 2 Report
The article bij Tarver et al. highlights an important dilemma in the management of anticoagulant therapy in patients with a history of mechanical heart valve placement and ischemic stroke. Although implantation of a mechanical heart valves requires lifelong anticoagulation with a vitamin K antagonist, adjustments in strategy are at times necessary. In this interesting case study a patient with a mechanical aortic valve presented himself at multiple occasions both with hemhorragic- and ischemic lesions, illustrating the difficult balance between protection from ischemic stroke and bleeding. After the third admission, the mechanical valve is replaced by a biological valve, eliminating the need for oral anticoagulation. The others continue by proposing an interesting (known) mechanism for the observed case, namely hemorrhagic conversion as a consequence from ischemic stroke.
Before publication I think the following points should be addressed:
Significant improvement in English writing are required. Examples:
Page 1: “Consequently, MHV require patients to under lifelong anticoagulant… “
Page 2: …”Vietnamese man initially presented with a 5 day history…”
Page 3: …at lunchtime and continued to worsen into the night.”
Page 4: “In the present case, the multi-disciplinary team withheld anticoagulation…”
Please write out abbreviations first (GRE, ECM, MMP)
Is a mechanical aortic valve the only medical history worth mentioning in this patient? Were there any other comorbidities/risk factors/family history? Were diagnostic tests performed to investigate any underlying (additional) anticoagulation disorders?
Is there any more information on the INR level stability in this patient? Upon admission the INR is 5.6, what was the INR upon the third admission? Is there any information available on the patient’s adherence to Coumadin? Is it possible that the cerebral hemorrhage was simply caused by excessively high INR levels, as opposed to hemorrhagic transformation?
The explanation as to why a mechanical heart valve was preferred over a biological valve and why no re-operation was performed after the second admission is of importance and should be mentioned on page 3 instead of in the final paragraph of the discussion section.
The paragraph on page 4 (“The formation of a thrombus on the surface…with the associated MRI findings (Figure 2)” is redundant.
The relevance of the paragraph describing the three main classes of MHV (page 5) is of little relevance.
What evidence was there that the mechanical valve was the source of thromboembolism?
Was trans-esophageal echocardiography performed to confirm a diagnosis of aortic valve thrombosis?
Did PA of the mechanical valve suggest valvular thrombosis?
It seems reasonable that hemorrhagic transformation would most likely only occur in post-infarction areas. Do the infarcted areas correspond with the locations of hemorrhage?
Author Response
Dear Sir or Madam,
Thank you for your comments and giving us the opportunity to improve our case report. Below is a list of the improvements made:
● Significant improvement in English writing are required. Examples:
○ Page 1: “Consequently, MHV require patients to under lifelong anticoagulant… “
○ Page 2: …”Vietnamese man initially presented with a 5 day history…”
○ Page 3: …at lunchtime and continued to worsen into the night.”
○ Page 4: “In the present case, the multi-disciplinary team withheld anticoagulation…”
■ Edits were made to correct grammatical errors and changes are highlighted in red
● Please write out abbreviations first (GRE, ECM, MMP)
○ The above abbreviations have been out written out first.
● Is a mechanical aortic valve the only medical history worth mentioning in this patient? Were there any other comorbidities/risk factors/family history? Were diagnostic tests performed to investigate any underlying (additional) anticoagulation disorders?
○ The only other significant medical history is patient’s Type 2 diabetes mellitus, no other relevant comorbidities, risk factors or family history. Patient was not hypertensive. A hypercoagulation panel was drawn, showing no evidence of hypercoagulability.
● Is there any more information on the INR level stability in this patient? Upon admission the INR is 5.6, what was the INR upon the third admission? Is there any information available on the patient’s adherence to Coumadin? Is it possible that the cerebral hemorrhage was simply caused by excessively high INR levels, as opposed to hemorrhagic transformation?
○ Yes, coumadin was held temporarily upon the observation that his INR was 5.6 at first admission. Patient was monitored with serial INRs, and his INR values dropped appropriately when the coumadin was temporarily held. INR dropped into the normal range and then was subtherapeutic/low normal [1.1-2.8] for over three months prior to the third hemorrhagic event.
● The explanation as to why a mechanical heart valve was preferred over a biological valve and why no re-operation was performed after the second admission is of importance and should be mentioned on page 3 instead of in the final paragraph of the discussion section.
○ The MHV vs bioprosthetic valve and its importance relative to the recommendation to forgo the redo surgery were placed in the 3rd paragraph as requested.
● The paragraph on page 4 (“The formation of a thrombus on the surface…with the associated MRI findings (Figure 2)” is redundant.]
○ Propensity and significance of thrombogenesis at the location of the MHV is important as it is the reason for valvular replacement after the 3rd encounter. However, the paragraph was removed per request.
● What evidence was there that the mechanical valve was the source of thromboembolism? Ultrasound or otherwise?
○ In this case report, the authors hypothesize that the areas of infarction and hemorrhage are due to microemboli that formed due to the presence of a mechanical aortic valve. It is known that clots may form on the surface mechanical valves, especially in the absence of anticoagulation. Our hypothesis is based primarily on radiographic findings. GRE T2-weighted 3-Tesla MRI demonstrated multiple hypointense lesions in multiple vascular territories, providing strong evidence for a microembolic pathophysiology that is most likely explained by the presence of a foreign mechanical valve.
● Was trans-esophageal echocardiography performed to confirm a diagnosis of aortic valve thrombosis?
○ Transesophageal echocardiography was not utilized until after the replacement of the mechanical valve with the biologic valve, simply to determine the integrity of the new biologic valve. Transthoracic echocardiography was performed upon the patient’s first hemorrhagic event; this study showed no evidence of thrombosis. Heart thrombosis would be a serious, life-threatening finding. Our study deals only with the ramifications of microthrombi formation which then embolize and cause small areas of infarction in multiple vascular territories throughout the cerebrum. These microthrombi are never seen grossly on echocardiography. However, high-power 3 Tesla magnets enable us to construct MRI images with fine detail of soft tissue structures that display these small areas of infarction and hemorrhage suggestive of microemboli.
● Did PA of the mechanical valve suggest valvular thrombosis?
○ We are unsure what the meaning is of the acronym “PA”. However, as stated above, the diagnosis of microemboli is more of a radiographic diagnosis rather than sonographic due to its microscopic nature. There were no gross signs of thrombosis on imaging, only the evidence of microemboli seen on finely detailed MRI imaging with a high-strength 3-Tesla magnet to enable adequate visualization of soft tissues. Given the clinical course, MRI findings, presence of a mechanical aortic valve, and intermittent holding of anticoagulation with 3 months of subtherapeutic/low normal INR values in this patient, the hypothesis of microemboli forming on and spreading from the MHV is well supported.
● It seems reasonable that hemorrhagic transformation would most likely only occur in post-infarction areas. Do the infarcted areas correspond with the locations of hemorrhage?
○ Yes, the infarcted areas do correspond with the locations of hemorrhage. The hypointense regions of hemorrhage seen on GRE T2-weighted 3-Tesla MRI imaging do in fact correspond directly to the contrast-enhancing areas on T1 post-contrast imaging with gadolinium. This congruity suggests endothelial cell dysfunction and proposes a mechanism to explain the observed hemorrhage. GRE hypointensities represent focal infarction and hemorrhage, whereas contrast-enhancing regions on T1 post-contrast imaging represent membrane permeability with gadolinium extravasation into the extracellular space. Since these hypointensities and contrast-enhancing regions are congruent, it is reasonable to conclude that ischemia and infarction was followed by endothelial cell dysfunction, increased vascular membrane permeability, and extravasation of red blood cells, yielding hemorrhage as observed on imaging. Due to restrictions imposed on case report submissions, not all images were included in our report. However, we have added one of the T1 post-contrast images to our revised draft which demonstrates this correlation between imaging studies.
Reviewer 3 Report
This is an interesting case but this needs a thorough work-up. Although the focus of the paper is on hemorrhagic transformation of emboli, several major points are missing:
- how was the variability of the INR (this is of special interest since the valve was placed in Vietnam)
- related to it: does the patient live in the US and how was the anticoagulation monitored
- has a prosthetic valve thrombosis been documented on ultrasound or otherwise
- have other sources of emboli been ruled out (such as chronic or paroxysmal atrial fibrillation, left ventricle in case of low LVEF)
- have other sources (hypercoagulability such as Leiden factor V mutations, chronic inflammatory status) been eliminated
- has the patient hypertension (as risk factor for intracerebral hemorrhage)
There is a lot of "common knowledge", in the paper. This could be omitted
Author Response
Dear Sir or Madam,
Thank you for your comments and giving us the opportunity to improve our case report. Below is a list of the improvements made:
● how was the variability of the INR (this is of special interest since the valve was placed in Vietnam)
○ INR was elevated to 5.6 on admission. Therefore, coumadin was held temporarily to decrease the INR into a safer range less dangerous for hemorrhage. Patient was monitored with serial INRs, and his INR values dropped appropriately when the coumadin was temporarily held. INR dropped into the normal range and then was subtherapeutic/low normal [1.1-2.8] for over three months prior to the valve replacement surgery.
● related to it: does the patient live in the US and how was the anticoagulation monitored
○ Yes, patient lives in the US now, although he is from Vietnam.
○ To monitor anticoagulation, coagulation profile was taken and PT/INR/PTT were drawn and measured regularly. PT was consistently elevated, and INR was subtherapeutic to low-normal [1.1-2.8] for over three months prior to the valve replacement surgery.
● has a prosthetic valve thrombosis been documented on ultrasound or otherwise
○ No, thrombosis has not been documented. Transthoracic echocardiography was performed upon the patient’s first hemorrhagic event; this study showed no evidence of thrombosis. Heart thrombosis would be a serious, life-threatening finding. Our study deals only with the ramifications of microthrombi formation which then embolize and cause small areas of infarction in multiple vascular territories throughout the cerebrum. These microthrombi are never seen grossly on echocardiography. However, high-power 3-Tesla magnets enable us to construct MRI images with fine detail of soft tissue structures that display these small areas of infarction and hemorrhage suggestive of microemboli.
● have other sources of emboli been ruled out (such as chronic or paroxysmal atrial fibrillation, left ventricle in case of low LVEF)
○ At each admission due to hemorrhagic events, patient underwent EKGs and telemetry, all of which showed normal results with no signs of atrial fibrillation or any type of arrhythmia. The transthoracic echocardiogram from the first hemorrhagic event reported a normal LVEF of 70%.
● have other sources (hypercoagulability such as Leiden factor V mutations, chronic inflammatory status) been eliminated
○ A hypercoagulability panel was performed which came back normal with no signs of hypercoagulability, including normal assay results for von Willebrand factor and factors V, VIII, X, and lupus anticoagulant was not detected. PT was elevated, and phospholipid-dependent screening tests (PT, DRVVT) were not prolonged. All of these labs rule out various differential diagnoses for hypercoagulability, increasing the likelihood that the presence of a mechanical aortic valve is the primary etiology of microemboli and innumerable areas of cerebral infarction in this patient.
○ We are unable to address this patient’s chronic inflammatory status directly since no CRP was drawn. However, upon his second hemorrhagic event, this patient underwent craniotomy with excision of the hematoma. The pathology report of this excised mass showed only a “blood clot with scant fragments of brain parenchyma with ischemic changes; no evidence of vascular malformation.”
● has the patient hypertension (as risk factor for intracerebral hemorrhage)
○ Patient does not have any history of hypertension.
Round 2
Reviewer 2 Report
The authors have made considerable changes to the manuscript which I feel have significantly improved the article.
2 remarks:
1. "Transthoracic echocardiogram done at this admission which showed a 70% LVEF with no evidence of thrombosis." Was also gradient across the M-valve measured, and the mitral valve area (MVA) to detect any form of stenosis? This relevant, in particular in view of the suboptimal INR levels. In view of the clinical picture, in our center TOE or CT would definitely be performed to judge valve function. In fact, I cannot image that a apparantly normal functioning valve is replaced without proper cardiological workup.
2. Conclusion: "Due to the paucity of therapeutic guidelines available and difficulty of randomized trial initiation within this population it is hoped that this case will add to the existing literature thereby helping clinicians treat this complex patient population." However, I do miss a clear recommendation for clinical practice. As I see it, this is just reflecting the risk of oral anticoagulation and mechanical heart valves, but it would not change my choice for the next similar patient of 43 years old. Or do the authors want to advocate a liberal re-operation of the valve in such cases?
Author Response
Dear Sir or Madam,
Thank you for your comments and giving us another opportunity to improve our case report. Below is a list of the additional improvements made:
1. "Transthoracic echocardiogram done at this admission which showed a 70% LVEF with no evidence of thrombosis."
· Was also gradient across the M-valve measured, and the mitral valve area (MVA) to detect any form of stenosis? Below is the reported Mitral valve values from the TTE on admission:
Please see attached .pdf file below for values
· This relevant, in particular in view of the suboptimal INR levels. In view of the clinical picture, in our center TOE or CT would definitely be performed to judge valve function. In fact, I cannot image that an apparently normal functioning valve is replaced without proper cardiological workup. Thank you for your insightful comments. Cardiology and Cardiothoracic surgery were both very active team members in the care of this complex patient. Many multi-disciplinary conferences were used to discuss the final treatment plan. Ultimately the decision to change the valve from a mechanical valve to a tissue valve was made so that the patient would not need to be on anticoagulants. After the valve was changed to the tissue valve only anti-platelet therapy was needed and fortunately they patient has had no further ischemic or hemorrhagic events.
2. Conclusion: "Due to the paucity of therapeutic guidelines available and difficulty of randomized trial initiation within this population it is hoped that this case will add to the existing literature thereby helping clinicians treat this complex patient population."
· However, I do miss a clear recommendation for clinical practice. As I see it, this is just reflecting the risk of oral anticoagulation and mechanical heart valves, but it would not change my choice for the next similar patient of 43 years old. Or do the authors want to advocate a liberal re-operation of the valve in such cases? Thank you for your comment. We have removed the sentence to avoid confusion and modified the first sentence of our conclusion. The authors do not recommend a liberal reoperation for heart valves but as reflected in our patient with multiple recurrent strokes it should be considered and can be done with a good outcome. Specifically, this should be considered when the risk of anticoagulation is greater than the clinical benefit and therefore a tissue valve with an antiplatelet regimen may be a safer option.

Reviewer 3 Report
Conclsions: change "definitive therapeutic pathway" into "possible therapeutic pathway"
Author Response
Dear Sir or Madam,
Thank you for your comments and giving us another opportunity to improve our case report. Below is the additional improvement made:
1. Conclusions: change "definitive therapeutic pathway" into "possible therapeutic pathway"
The authors have changed the text accordingly. Thank you.
